# Multiplex PCR Assay for Simultaneous Identification of Five Types of Tuna (*Katsuwonus pelamis*, *Thunnus alalonga*, *T. albacares*, *T. obesus* and *T. thynnus*)

**DOI:** 10.3390/foods11030280

**Published:** 2022-01-20

**Authors:** Ga-Young Lee, Seung-Man Suh, Yu-Min Lee, Hae-Yeong Kim

**Affiliations:** Institute of Life Sciences & Resources, Department of Food Science & Biotechnology, Kyung Hee University, Yongin 17104, Korea; gayoung.lee0731@gmail.com (G.-Y.L.); teri2gogo@naver.com (S.-M.S.); lym5373@naver.com (Y.-M.L.)

**Keywords:** multiplex PCR, species identification, *Katsuwonus pelamis*, *Thunnus alalonga*, *T. albacares*, *T. obesus*, *T. thynnus*

## Abstract

There is a need to identify the species of similar types of fish, especially those that are commercially sold. Particularly, the price of tuna varies depending on its type, which is difficult to determine as they are sold in cut or processed forms. This study developed a multiplex polymerase chain reaction (PCR) assay to identify the five most common tuna species: bigeye, skipjack, Atlantic bluefin, albacore, and yellowfin tunas. Newly designed species-specific primer sets for these five tuna species were created. Subsequently, the amplicon sizes obtained were 270, 238, 200, 178, and 127 base pairs for bigeye, skipjack, Atlantic bluefin, albacore, and yellowfin tunas, respectively. Each primer’s specificity was further tested using 15 other fish species, and no cross-reactivity was observed. To identify multiple targets in a single reaction, multiplex PCR was optimized to increase its resolution and accuracy. The detection levels of the multiplex PCR assay were confirmed to be 1 pg for all the five tunas. Additionally, it was successfully applied to 32 types of commercial tuna products. Therefore, this multiplex PCR assay could be an efficient identification method for various tuna species.

## 1. Introduction

Recently, species identification in fish and fishery products has been a topic of increasing concern [1,2,3]. Seafood fraud occurs due to intentional fish species substitution and the incorrect labeling of fresh or processed fishery products. Tuna is one of the most popular fish [4] with a high consumption rate. However, since tuna is usually sold in vacuum-packaged pieces or canned form [5], its morphological characteristics cannot be distinguished, making accurate species identification difficult [6]. Additionally, the quality and price of each tuna species are different [4]. For this reason, fraudulent substitutions may occur with relatively inexpensive species (e.g., *Katsuwonus pelamis*) [1], which means that identification of the fish species is important to avoid fraudulent and vague labeling of canned tuna [7]. Therefore, analysis procedures for the identification of different tuna species are required.

Protein-based assays for fish species identification have been previously reported, such as isoelectric focusing, isoelectric focusing in immobilized pH gradients, two-dimensional electrophoresis, and enzyme-linked immunosorbent assay [8,9,10,11,12,13,14,15]. However, protein-based analytical methods for species identification are limited because of the proclivity of proteins to denature during heat and high-pressure treatment [16]. DNA-based methods can also be employed to characterize seafood species [17]. Compared to proteins, DNA is more stable during heat and high-pressure processing [18,19]. In addition, mitochondria remain after processing, which avoids DNA loss, and since mitochondrial DNA is more abundant than nuclear DNA, it is easier to detect [20,21]. Real-time polymerase chain reaction (PCR) [4,5,22,23,24], multiple PCR [19,21,25,26], loop-mediated isothermal amplification (LAMP) [6], PCR restriction fragment length polymorphism (PCR-RFLP) [7,27], and others have been reported as viable DNA-based PCR techniques for tuna species identification. Real-time PCR assays require an expensive machine and have the disadvantage that it must be done by skilled experimenters. On the other hand, multiplex PCR assays use a simple PCR machine and a gel-doc system available in most laboratories [28]. Additionally, in the classical PCR, template DNA of a single species is amplified in a single PCR run, requiring several runs to detect multiple target species, resulting in additional cost and time [29]. However, multiplex PCR amplifies multiple DNA templates simultaneously in a single reaction, making it both cost- and time-effective [30,31,32].

The most commercially available tuna species are *Katsuwonus pelamis*, *Thunnus alalonga*, *T. albacares*, *T. obesus*, and *T. thynnus*. Previous studies involving multiplex PCR targeting these species included multiplex primer-extension assays [25].

Therefore, this study aimed to develop a multiplex PCR assay for the rapid and sensitive detection of five commercially available tuna species.

## 2. Materials and Methods

### 2.1. Samples

The five species of tuna: bigeye *(T. obesus*), skipjack (*K. pelamis*), Atlantic bluefin (*T. thynnus*), albacore (*T. alalunga*), and yellowfin (*T. albacares*) and longtooth grouper (*Epinephelus bruneus*), convict grouper (*Epinephelus septemfasciatus*), marlin (*Tetrapturus angustirostris*), and sailfish (*Istiophorus platypterus*) were obtained from the Ministry of Food and Drug Safety (Osong, Korea) as reference samples. Eleven other nontarget seafood (fish) products such as common carp (*Cyprinus carpio*), goldfish (*Carassius auratus*), Chinese muddy loach (*Misgurnus mizolepis*), snakehead (*Channa argus*), Pacific cod (*Gadus macrocephalus*), Alaska pollock (*G. chalcogrammus*), Nile tilapia (*Oreochromis niloticus*), Pacific saury (*Cololabis saira*), Pacific chub mackerel (*Scomber japonicus*), Atlantic salmon (*Salmo salar*), and masu salmon (*Oncorhynchus masou*) were purchased from the online store and morphologically confirmed through the image database of the National Institute of Fisheries Science. Thirty-two processed food products were purchased from online shops in Korea as monitoring samples. All samples were stored at −20 °C prior to DNA extraction.

### 2.2. DNA Extraction

Fresh raw materials were used for specificity and limit of detection (LOD), and processed tuna products, including dried, heated, and canned products, were used as the monitoring samples. DNA was extracted from 25-mg ground samples using the DNeasy Tissue Kit (Qiagen, Hilden, Germany), as per the manufacturer’s instructions, except for from canned tuna, which was pretreated in water to remove oil and lipids. Then, DNA was extracted from the pretreated canned tuna using the cetyl trimethyl ammonium bromide method [32,33]. The extracted DNA’s concentration and purity were measured using a Maestro Nano Micro-Volume Spectrophotometer (Maestro, Las Vegas, NV, USA).

### 2.3. Species-Specific Multiplex Primer Design

The target genes of bigeye, skipjack, Atlantic bluefin, albacore, and yellowfin tunas were selected from their mitochondrial DNA sequences. The gene sequences of the five tuna species and the 15 nontarget species were obtained from the National Center for Biotechnology Information (NCBI, www.ncbi.nlm.nih.gov/, accessed on 21 August 2021) and aligned using the Clustal Omega program (http://www.ebi.ac.uk/Tools/msa/clustalo/, accessed on 21 August 2021). Primers were designed by the Primer Design program version 3.0 (Scientific and Educational Software, Durham, NC, USA) and synthesized by Bionics (Seoul, Korea); their sequences and details are listed in Table 1. The sequence alignment of the target gene and the position of the designed primer are shown in Appendix A.

### 2.4. Single and Multiplex PCR

Single PCR was conducted under the following conditions: the final volume was 25 µL, containing 2.5 µL 10× buffer (Bioneer, Daejeon, Korea), 0.2 mM dNTPs, 0.5 unit Hot Start *Taq* DNA polymerase (Bioneer), 0.4 µM of each primer, and 10 ng template DNA. Amplification was conducted in a thermal cycler (Model PC 808, ASTEC, Fukuoka, Japan) as follows: predenaturation at 95 °C for 5 min; 40 cycles of 95 °C for 30 s, 62 °C for 30 s, and 72 °C for 30 s; and a final extension at 72 °C for 5 min. The amplification products were electrophoresed on 2% agarose gel stained with ethidium bromide at 150 V for 12 min.

For multiplex PCR, the optimized concentration of primers (Table 1) and 1 unit of Hot Start *Taq* DNA polymerase were used, and the rest of the conditions were the same as single PCR. The amplification products from the multiplex PCR were electrophoresed on 3% agarose gel stained with ethidium bromide at 150 V for 30 min. Additionally, capillary electrophoresis was conducted using an Agilent 2100 Bioanalyzer (Agilent Technologies, Santa Clara, CA, USA) to confirm the sensitive accuracy of multiplex PCR.

## 3. Results and Discussion

### 3.1. Species-Specific Multiplex Primer Design

The mitochondrial DNA of Atlantic bluefin tuna was found to be ≥98% and 97% identical to that of yellowfin and bigeye tunas and albacore tuna, respectively. The degree of similarity was further confirmed by Basic Local Alignment Search Tool (BLAST) search on the NCBI database. Koenig and Ziebell (2014) developed a primer with a modified base sequence for the real-time PCR detection [34]. We develop a species-specific primer with a modified base sequence for Atlantic bluefin tuna.

### 3.2. Specificity and Sensitivity of Single PCR

To verify the designed primer’s specificity, single PCR was conducted with the DNA extracted from raw materials. The bigeye, skipjack, Atlantic bluefin, albacore, and yellowfin tuna primers produced amplification products of 270, 238, 200, 178, and 172 base pairs, respectively. The primers did not cross-react with the 15 nontarget species, including common carp, goldfish, Chinese muddy loach, snakehead, Pacific cod, Alaska pollock, Nile tilapia, Pacific saury, Pacific chub mackerel, longtooth grouper, convict grouper, Atlantic salmon, masu salmon, marlin, and sailfish (Figure 1).

Additionally, each primer underwent sensitivity tests. Primer sensitivity was confirmed by diluting the DNA of the target species by 10-fold, from 10 ng to 0.00001 ng. Skipjack tuna demonstrated a sensitivity of 1 pg, bigeye tuna of 0.1 pg, and albacore, yellowfin, and Atlantic bluefin tunas of 0.01 pg (Figure 2). Compared to previous study [35], the primers developed in this study could detect bigeye, skipjack, albacore, and yellowfin tunas’ DNA even in low concentrations. Conventional PCR has been reported to be less specific and sensitive than real-time PCR [35,36]. However, the assay conducted in this study based on conventional PCR had higher detection sensitivity than reported by previous studies. The sensitivities of bigeye, albacore, yellowfin, and skipjack tunas in the real-time PCR assay were 1.81, 1.68, 4.49, and 2.41 pg, respectively [35]. Therefore, the detection limit of the method used in this study was within 0.01 pg, which indicates that it is more sensitive than other methods.

### 3.3. Specificity and Sensitivity of Multiplex PCR

To determine the optimal concentration of multiplex PCR primer sets, we performed preliminary experiments (data not shown). As a result, we found that the multiplex PCR condition (i.e., 1.4 µM/1.2 µM/0.4 µM/0.6 µM/1.2 µM for bigeye tuna/skipjack tuna/Atlantic bluefin tuna/albacore tuna/yellowfin tuna, respectively) was most suitable considering the cross-reactivity of the primer sets and sensitivity. The multiplex PCR was optimized by setting the annealing temperature and time to 62 °C and 30 s, respectively, which is the same as single PCR. This optimized multiplex PCR was developed for the simultaneous identification of the five types of tuna. Additionally, the specificity of multiplex PCR was confirmed by gel electrophoresis. The target fish species was amplified by its specific primer and no nonspecific band or false-positive amplification of the negative control group was observed (Figure 3).

The sensitivity of multiplex PCR was confirmed by gel (Figure 4) and capillary electrophoreses (Figure 5) using diluted DNA (10 ng to 0.01 pg). Capillary electrophoresis has been reported to provide higher resolution than agarose-based electrophoresis when distinguishing DNA amplicons [36]. Then, the multiplex PCR method coupled with capillary electrophoresis was conducted to detect the five species of tunas [36]. The sensitivity of the developed multiplex PCR detected by capillary and agarose-based electrophoreses was 1 pg (Figure 4 and Figure 5). Similar studies have reported a sensitivity of 5 ng for tuna and billfish and 1 ng for blowfish and freshwater fish species [26,37,38]. Our assay demonstrates better sensitivity than those in previously reported studies. These results indicated that primers specific for the five tuna species did not cross-react with nontarget species and only amplified the target species, which enabled discrimination among the five similar species. Additionally, this assay displays a sensitivity of 1 pg, which provides accuracy and sensitivity in the detection and differentiation of the target species from the five tuna species. The detailed data on specificity, accuracy, and sensitivity calculated in accordance with ISO standards are shown in Appendix A.

### 3.4. Application and Validation toward Commercial Food Products

To evaluate the applicability of the multiplex PCR assay developed in this study, we employed it to identify the presence of the five species of tuna in 32 commercial tuna and food products (18 raw tunas, 12 tuna cans, and 2 dried tunas) (Table 2) by three different experimenters. The results obtained were compared to the species that was indicated on the label of the products to determine its correctness. As shown in Table 2, commercially available tuna and food products displayed contradictory results for tuna species when compared to what was indicated in their labels.

Five (numbers 28, 29, 30, 31, and 32) of the twelve cans had no tuna species identification on their labeling. Consequently, the product labeled “Light Tuna” comprised a mixture of several tuna species. Atlantic bluefin and albacore tunas were mixed in three of the five products (numbers 28, 29, and 31). Of the two other products, one (number 30) contained skipjack, Atlantic bluefin, and albacore tunas, and the other (number 32) included Atlantic bluefin, albacore, and yellowfin tunas. In the remaining seven cans (numbers 8, 9, 10, 11, 12, 22, and 23), the tuna content of only two products (numbers 22 and 23) matched their labels. Five cans (numbers 8, 9, 10, 11, and 12) were labeled with skipjack tuna, one (number 22) with albacore tuna, and the remaining one (number 23) with yellowfin tuna. Of the five skipjack-marked tuna cans, two contained yellowfin and albacore tunas (numbers 8 and 11, respectively) and two (numbers 9 and 12) contained a mixture of Atlantic bluefin and albacore tunas. The remaining one (number 10) was a mixture of skipjack, Atlantic bluefin, and albacore tunas. Of the 18 raw tunas, 14 amplified a species that matched their respective labels and 4 (numbers 2, 6, 24, and 26) amplified a species that did not match their respective labels. Of the two products (numbers 2 and 6) marked with bigeye tuna, one (number 2) amplified yellowfin tuna and the other (number 6) amplified the Atlantic bluefin tuna, and the two products (number 24 and 26) marked as yellowfin tuna amplified the Atlantic bluefin tuna. Dried tunas (numbers 7 and 17) amplified a species that did not match their corresponding labeling. The product marked as dried skipjack tuna (number 7) contained a mixture of skipjack, Atlantic bluefin, and yellowfin tunas. It was also confirmed that the dried product (number 17) marked as Atlantic bluefin tuna contained albacore tuna.

To identify the PCR amplicons of commercial samples that were inconsistent with their labels, a single PCR was conducted with each primer of the amplified target. The PCR amplicons were then isolated and sequenced. The results of the single PCR were consistent with those of the multiplex PCR (Table 2 and Figure 6). Furthermore, the nucleotide sequences of the PCR product were compared with BLAST nucleotide sequences in the NCBI database (data not shown). The sequencing results were identical to those shown in Table 2, indicating that a few commercial samples were mislabeled or mixed with species other than what their labels specified.

Servusova and Piskata (2021) tested 70 samples and reported 12 products to be mislabeled, and 2 products to be a mixture of two species [22]. Bojolly et al. (2017) tested 10 commercial products labeled with yellowfin tuna, five of which were found to be a mixture of yellowfin and skipjack tunas [24]. Substitutions among tuna species occur during the filleting and can manufacturing processes, leading to inaccurate labels [24]. Additionally, the identification of the species is difficult due to processes such as sterilization, and hence, species other than the marked species may be present. In addition, some Atlantic populations of *T. thynnus* might exhibit introgression of mitochondrial DNA from *T. alalonga* [39]. Therefore, some mislabeling involving Atlantic bluefin and albacore tunas could remain undetected [40]. However, with single and multiplex specificity verification, the primers developed in this study can be used as genetic markers for tuna species identification.

## 4. Conclusions

We designed five primer sets to develop the multiplex PCR for simultaneously distinguishing and identifying five tuna species (bigeye, skipjack, Atlantic bluefin, albacore, and yellowfin tunas). Primer sets for multiplex PCR assay were species-specific and sensitive with a 1 pg detection limit. The applicability test accurately identified the types of tuna in commercial tuna products. Hence, this multiplex PCR assay can be employed to confirm the validity of the labels of highly degraded, commercially processed foods quickly and at low cost.

## Figures and Tables

**Figure 1 foods-11-00280-f001:**
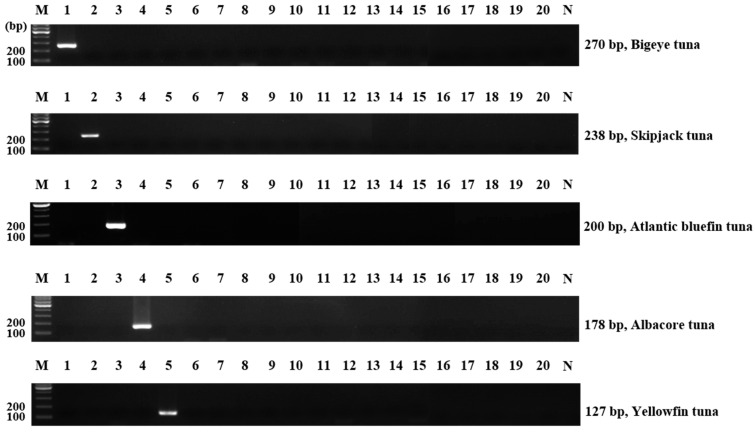
Specificity of single PCR using newly designed primers. Lane M, 100-bp DNA ladder; lanes 1–23, bigeye tuna, skipjack tuna, Atlantic bluefin tuna, albacore tuna, yellowfin tuna, common carp, goldfish, Chinese muddy loach, snakehead, Pacific cod, Alaska pollock, Nile tilapia, Pacific saury, Pacific chub mackerel, longtooth grouper, convict grouper, Atlantic salmon, masu salmon, marlin, sailfish; lane N, non-template.

**Figure 2 foods-11-00280-f002:**
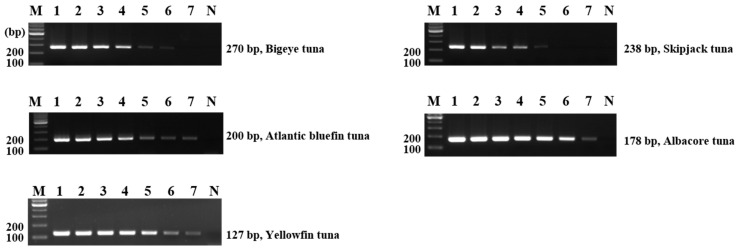
Sensitivity of single PCR using newly designed primers. Lane M, 100-bp DNA ladder; lanes 1–7, positive gDNA 10, 1, 0.1, 0.01, 0.001, 0.0001, 0.00001 ng; lane N, non-template.

**Figure 3 foods-11-00280-f003:**
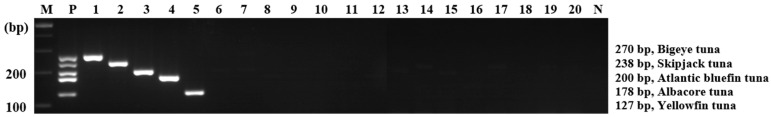
Specificity of multiplex PCR. Lane M, 100-bp DNA ladder; lane P, positive control (10 ng of DNA from target species); lanes 1–20, bigeye tuna, skipjack tuna, Atlantic bluefin tuna, albacore tuna, yellowfin tuna, common carp, goldfish, Chinese muddy loach, snakehead, Pacific cod, Alaska pollock, Nile tilapia, Pacific saury, Pacific chub mackerel, longtooth grouper, convict grouper, Atlantic salmon, masu salmon, marlin, sailfish; lane N, non-template.

**Figure 4 foods-11-00280-f004:**
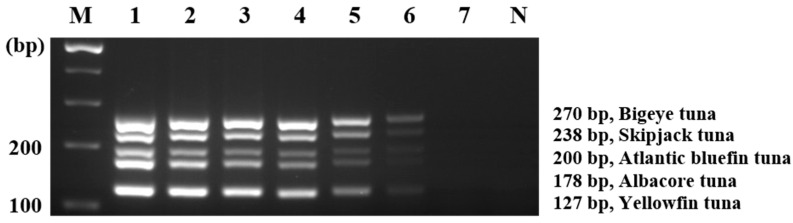
Sensitivity of multiplex PCR. Lane M, 100-bp DNA ladder; lanes 1–7, positive gDNA 10, 1, 0.1, 0.01, 0.001, 0.0001, 0.00001 ng; lane N, non-template.

**Figure 5 foods-11-00280-f005:**
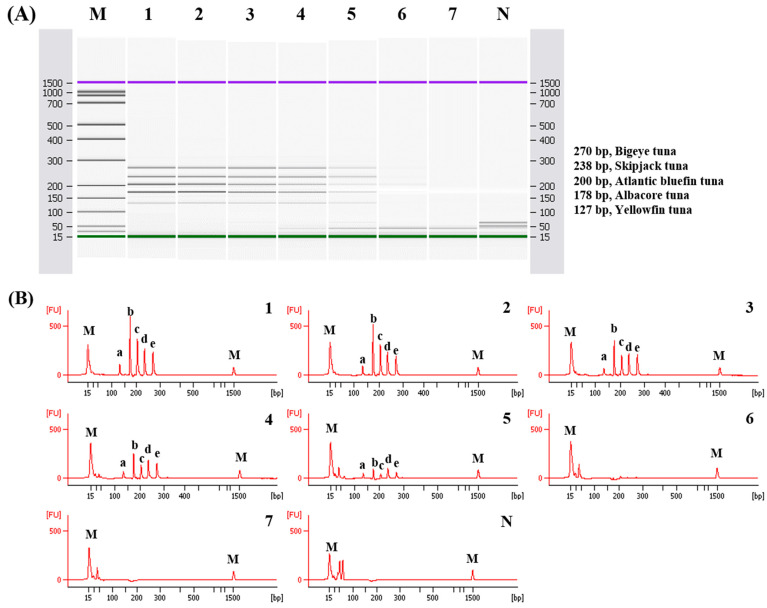
Electropherograms of sensitivity results of the multiplex PCR assay. Gel image (**A**) and electropherograms (**B**). FU, fluorescence; M, alignment marker; lanes 1–7, positive gDNA 10, 1, 0.1, 0.01, 0.001, 0.0001, 0.00001 ng; lane N, non-template; a,b,c,d, and e indicate yellowfin tuna, albacore tuna, Atlantic bluefin tuna, skipjack tuna, bigeye tuna respectively.

**Figure 6 foods-11-00280-f006:**
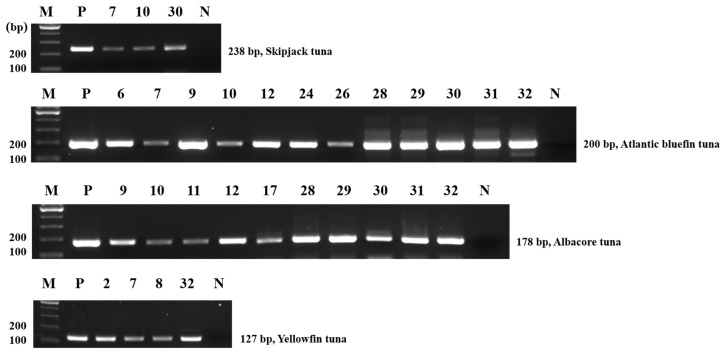
Single PCR of the adulterated commercial sample. Lane M, 100 bp DNA ladder; lane P, positive control (10 ng of DNA from target species); lane N, non-template; the lane number matches the monitoring number in Table 2.

**Table 1 foods-11-00280-t001:** Primers used in the multiplex PCR.

Target Species	Target Gene	Primer Name	Sequence (5′ → 3′)	Amplicon Size (bp)	Concentration (μM)	Accession No.	Reference
*Thunnus* *obesus*	ATP6	Obe-F	ACT TGC ATT CCC CCT ATG G	270	1.4	KY400011.1	This study
		Obe-R	GCT GTT AGG ATT GCC ACA G				
*Katsuwonus pelamis*	Cytb	Kat-F	GGT CCT AGC TCT TCT TGC A	238	1.2	NC_005316.1	This study
		Kat-R	TGC AAG TGG GAA GAA GAT G				
*Thunnus thynnus*	NADH5	Thy-F	AAC TCT TTA TCG GGT GGG AG	200	0.4	KF906720.1	This study
		Thy-R	^1^ AGC GGT TAC GAA CAT TTG C**T**T C				
*Thunnus* *alalunga*	Cytb	Ala-F	GTT TCG TGA TCC TGC TAG TG	178	0.6	NC_005317.1	This study
		Ala-R	CCT CCT AGT TTG TTG GAA TAG AT				
*Thunnus* *albacares*	NADH4	Alba-F	CAT GAT TGC CCA CGG ACT TA	127	1.2	KM588080.1	This study
		Alba-R	TGT TGT TAT AAG GGG CAG C				

^1^ The nucleotide sequence G was replaced with T (in bold).

**Table 2 foods-11-00280-t002:** Application and intra-laboratory validation results of the multiplex PCR assay to commercial tuna products.

No	Product Type	Labeled Species		Multiplex PCR Results			
			Bigeye Tuna	Skipjack Tuna	Atlantic Bluefin Tuna	Albacore Tuna	Yellowfin Tuna
1	Raw	Bigeye tuna	+++				
2	Raw	Bigeye tuna					+++
3	Raw	Bigeye tuna	+++				
4	Raw	Bigeye tuna	+++				
5	Raw	Bigeye tuna	+++				
6	Raw	Bigeye tuna			+++		
7	Dried	Skipjack tuna		+++	+++		+++
8	Canned	Skipjack tuna					+++
9	Canned	Skipjack tuna			+++	+++	
10	Canned	Skipjack tuna		+++	+++	+++	
11	Canned	Skipjack tuna				+++	
12	Canned	Skipjack tuna			+++	+++	
13	Raw	Skipjack tuna		+++			
14	Raw	Skipjack tuna		+++			
15	Raw	Skipjack tuna		+++			
16	Raw	Skipjack tuna		+++			
17	Dried	Atlantic bluefin tuna				+++	
18	Raw	Atlantic bluefin tuna			+++		
19	Raw	Atlantic bluefin tuna			+++		
20	Raw	Atlantic bluefin tuna			+++		
21	Raw	Atlantic bluefin tuna			+++		
22	Canned	Albacore tuna				+++	
23	Canned	Yellowfin tuna					+++
24	Raw	Yellowfin tuna			+++		
25	Raw	Yellowfin tuna					+++
26	Raw	Yellowfin tuna			+++		
27	Raw	Yellowfin tuna					+++
28	Canned	Light Tuna			+++	+++	
29	Canned	Light Tuna			+++	+++	
30	Canned	Light Tuna		+++	+++	+++	
31	Canned	Light Tuna			+++	+++	
32	Canned	Light Tuna			+++	+++	+++

‘+’ means a positive result.

## Data Availability

The data presented in this study are available on request from the corresponding author.

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
