# Peer review of "Multiplex PCR Assay for Simultaneous Identification of Five Types of Tuna (Katsuwonus pelamis, Thunnus alalonga, T. albacares, T. obesus and T. thynnus)"

_foods, 2022, doi:10.3390/foods11030280_

Round 1

Reviewer 1 Report

This study describes an useful, simple methodology based on a cocktail of species-specific primers (Pentaplex primer design) for the identification of five tuna species. These species are the most commercially important tunas, thus the article is timely and relevant. 

The methodology proposed is technically sound. The authors demonstrate it is sensitive and specific.

I only see a possible problem regarding the last point in Thunnus thynnus and T. alalunga. Some Atlantic populations of T. thynnus exhibit introgression of mitochondrial DNA from T. alalunga (Alvarado Bremer et al. 2005). Therefore, some mislabelling involving Atlantic T. thynnus and T. alalunga could remain undetected based on mitochondrial genetic markers (Viñas & Tudela 2009), as it is the case in this article. Since this affects two of the five species tested here, the issue should be mentioned in the Discussion as a possible limitation of the methodology. 

The articles mentioned are:

Alvarado Bremer JR, Viñas J, Mejuto J, Ely B, Pla C. (2005) Comparative phylogeography of Atlantic bluefin tuna and swordfish: the combined effects of vicariance, secondary contact, introgression, and population expansion on the regional phylogenies of two highly migratory pelagic fishes. Molecular Phylogenetics and Evolution 36(1):169-187. https://doi.org/10.1016/j.ympev.2004.12.011.

Viñas, J., Tudela, S. (2009). A Validated Methodology for Genetic Identification of Tuna Species (Genus Thunnus). PLoS ONE, 4(10), e7606. 

Author Response

This study describes an useful, simple methodology based on a cocktail of species-specific primers (Pentaplex primer design) for the identification of five tuna species. These species are the most commercially important tunas, thus the article is timely and relevant. 

The methodology proposed is technically sound. The authors demonstrate it is sensitive and specific.

I only see a possible problem regarding the last point in Thunnus thynnus and T. alalunga. Some Atlantic populations of T. thynnus exhibit introgression of mitochondrial DNA from T. alalunga (Alvarado Bremer et al. 2005). Therefore, some mislabelling involving Atlantic T. thynnus and T. alaldunga could remain undetected based on mitochondrial genetic markers (Viñas & Tudela 2009), as it is the case in this article. Since this affects two of the five species tested here, the issue should be mentioned in the Discussion as a possible limitation of the methodology. 

Response: As you recommended, we revised the sentence in lines 255-260 in the discussion as follows:

Lines 255-260: In addition, some Atlantic populations of T. thynnus might exhibit introgression of mitochondrial DNA from T. alalunga [39]. Therefore, some mislabeling involving Atlantic bluefin and albacore tunas could remain undetected [40]. However, with single and multiplex specificity verification, the primers developed in this study can be used as genetic markers for tuna species identification.

Reviewer 2 Report

Overall, I’m a bit confused as to why you created this assay when qPCR markers for these species already exist? Is this assay meant to improve on that method in some way? Or are you suggesting that this method could be useful for anyone concerned with tuna products that doesn’t have access to qPCR equipment? The work you’ve done seems well developed, but I’m just not sure there is a need for it. You should add some text to the Introduction and Conclusion to explain why the development of this particular assay is important and useful to the broader community.

Other minor comments:

Title: Re-order the species names into alphabetical order and then abbreviate Thunnus to T. for 2nd through 4th species: (Katsuwonus pelamis, Thunnus alalunga, T. albacares, T. obesus, and T. thynnus). Also the ‘and’ between the species names should not be italicized.

Probably makes more sense to use ‘Multiplex’ over ‘Pentaplex’ throughout the manuscript

Line 15: change ‘procured’ to ‘created’

Line 27: Why are ‘fish’ and ‘fishery products’ in italics?

Line 71: Define LOD before using the acronym

Line 99-100: Is the Taq also from Bioneer?

Line 105: How were the primer concentrations optimized? Just add ‘(Table 1)’ after ‘concentration of primers’ and then delete the later sentence (line 107) ‘The concentrations of the five primers are shown in Table 1).’

Line 117-119: I don’t understand what this sentence about potato/virus primers has to do with this paper.

Line 119-121: I’m not sure why you’re explaining this. Is the original primer from some other reference? If not, there doesn’t seem to be a reason to report the ‘original’ primer sequence if the new one performs better.

Line 235 & 236: Delete ‘reaction’ after ‘PCR’

Author Response

Overall, I’m a bit confused as to why you created this assay when qPCR markers for these species already exist? Is this assay meant to improve on that method in some way? Or are you suggesting that this method could be useful for anyone concerned with tuna products that doesn’t have access to qPCR equipment? The work you’ve done seems well developed, but I’m just not sure there is a need for it. You should add some text to the Introduction and Conclusion to explain why the development of this particular assay is important and useful to the broader community.

Response: Real-time PCR assays require an expensive machine and have the disadvantage that it must be done by skilled experimenters. On the other hand, multiplex PCR assays use a simple PCR machine and a gel-doc system available in most laboratories [28]. Additionally, in the classical PCR, template DNA of a single species is amplified in a single PCR run, requiring several runs to detect multiple target species, resulting in additional cost and time [29]. However, multiplex PCR amplifies multiple DNA templates simultaneously in a single reaction, making it both cost- and time-effective [30-32]. As you said the qPCR markers already exist, but the markers developed in this study are more sensitive, as mentioned in lines 142-145. Also, these markers can simultaneously detect five species and have a sensitivity of 1 pg.

As you recommended, we revised the sentence in lines 49-54 and 269-271 in the Introduction and Conclusion as follows:

Lines 49-54: Real-time PCR assays require an expensive machine and have the disadvantage that it must be done by skilled experimenters. On the other hand, multiplex PCR assays use a simple PCR machine and a gel-doc system available in most laboratories [28]. Additionally, in the classical PCR, template DNA of a single species is amplified in a single PCR run, requiring several runs to detect multiple target species, resulting in additional cost and time [29].

Lines 269-271: Hence, this multiplex PCR assay can be employed to confirm the validity of the labels of highly degraded, commercially processed foods quickly and at low cost.

Other minor comments:

Title: Re-order the species names into alphabetical order and then abbreviate Thunnus to T. for 2nd through 4th species: (Katsuwonus pelamis, Thunnus alalunga, T. albacares, T. obesus, and T. thynnus). Also the ‘and’ between the species names should not be italicized.

Response: As you recommended, we revised title, keywords, and the sentence in lines 56-57 as follows:

Lines 56-57: The most commercially available tuna species are Katsuwonus pelamis, Thunnus alalonga, T. albacares, T. obesus, and T. thynnus.

Probably makes more sense to use ‘Multiplex’ over ‘Pentaplex’ throughout the manuscript

Response: As you recommended, we revised ‘Pentaplex’ to ‘Multiplex’ throughout the manuscript.

Line 15: change ‘procured’ to ‘created’

Response: As you recommended, we revised ‘procured’ to ‘created’ in line 15.

Line 27: Why are ‘fish’ and ‘fishery products’ in italics?

Response: Thank you for your comment. We changed italics to regular font for ‘fish’ and ‘fishery products’ in lines 26-27 as follows:

Lines 26-27: Recently, species identification in fish and fishery products has been a topic of increasing concern [1-3].

Line 71: Define LOD before using the acronym

Response: We defined ‘LOD’ to ‘limit of detection (LOD)’ in line 77.

Line 99-100: Is the Taq also from Bioneer?

Response: As you recommended, we revised the sentence in line 101 as follows:

Line 101: 0.5 unit Hot Start Taq DNA polymerase (Bioneer)

Line 105: How were the primer concentrations optimized?

Response: Thank you for your comments. Competition between primer sets in multiplex PCR may affect its specificity and sensitivity. Previous multiplex PCR studies optimized concentration of primers to increase sensitivity and minimize non-specific interactions [Cheng et al, 2016].

Likewise, we adjusted the concentration of each primer set to optimize the multiplex PCR conditions. Preliminary experiments were performed to determine the optimal concentration of the primer sets. The primer concentration (μM) of bigeye tuna/skipjack tuna/Atlantic bluefin tuna/albacore tuna/yellowfin tuna was as follows;

1) 0.4/0.4/0.4/0.4/0.4, 2) 0.8/0.8/0.4/0.4/0.6, 3) 1.2/0.8/0.4/0.4/0.8, 4) 1.4/1.2/0.4/0.6/1.2, and 5) 1.4/1.2/0.4/0.4/1.4. As a result of testing with various concentrations, the primer concentration of 4) 1.4/1.2/0.4/0.6/1.2 was most suitable considering the cross-reactivity of the primer sets and sensitivity.

Cheng, F.; Wu, J.; Zhang, J.; Pan, A.; Quan, S.; Zhan, D.; Kim, H.Y.; Li, X.; Zhou, S.; Yang, L. Development and inter-laboratory transfer of a decaplex polymerase chain reaction assay combined with capillary electrophoresis for the simultaneous detection of ten food allergens. Food Chem. 2016, 199, 799–808.

As you recommended, we revised the sentence in lines 151-157 as follows:

Lines 151-157: To determine the optimal concentration of multiplex PCR primer sets, we performed preliminary experiments (data not shown). As a result, we found that the multiplex PCR condition (i.e., 1.4 µM/1.2 µM/0.4 µM/0.6 µM/1.2 µM for bigeye tuna/skipjack tuna/Atlantic bluefin tuna/albacore tuna/yellowfin tuna, respectively) was most suitable considering the cross-reactivity of the primer sets and sensitivity. The multiple PCR was optimized by setting the annealing temperature and time to 62°C and 30 s, respectively, which is the same as single PCR.

Just add ‘(Table 1)’ after ‘concentration of primers’ and then delete the later sentence (line 107) ‘The concentrations of the five primers are shown in Table 1.’

As you recommended, we deleted the sentence (line 107) and added ‘(Table 1)’ in line 106 as follows:

Line 106: For multiplex PCR, the optimized concentration of primers (Table 1) and 1 unit of Hot Start Taq DNA polymerase were used, and the rest of the conditions were the same as single PCR.

Line 117-119: I don’t understand what this sentence about potato/virus primers has to do with this paper.

Response: As you recommended, we revised the sentence in lines 117-118 as follows:

Lines 117-118: Koenig & Ziebell (2014) developed a primer with a modified base sequence for the real-time PCR detection [34].

Lines 119-121: I’m not sure why you’re explaining this. Is the original primer from some other reference? If not, there doesn’t seem to be a reason to report the ‘original’ primer sequence if the new one performs better.

Response: The five species-specific primers in this study were newly developed.

As you recommended, we revised the sentence in lines 118-119 as follows:

Lines 118-119: We develop a species-specific primer with a modified base sequence for Atlantic bluefin tuna.

Line 235 & 236: Delete ‘reaction’ after ‘PCR’

Response: As you recommended, we deleted ‘reaction’ after ‘PCR’ in lines 236-239. 

Reviewer 3 Report

This is an interesting paper that focuses on food frauds. In particular, the study aims to develop a pentaplex PCR assay for the rapid detection of five commercially available tuna species, against fraudulent species substitution in fishery products.

The topic is current, the paper is pretty well written and aims and methods are sufficiently reported. However, there are some lacks in the reference section and minor revisions are needed:

L 30-32: Please, add some references about tuna species identification against frauds

36: Also, two-dimensional electrophoresis and mass spectrometric techniques were used, please add them and cite:

Pepe T., Ceruso M., Carpentieri A., Ventrone I., Amoresano A., Anastasio A. (2010) “Proteomic analysis for the identification of Thunnus genus three species” Vet Res Commun, 34 (Suppl 1):S153–S155

L 57-67: Please, specify how did you proceed with the species recognition and who did it (morphological? From expert?)

Figure 3: Please, specify which is “P” in the legend

Author Response

L 30-32: Please, add some references about tuna species identification against frauds

Response: Thank you for your comment. We added references [1,7] and the sentence in lines 32-35 as follows:

Lines 32-35: For this reason, fraudulent substitutions may occur with relatively inexpensive species (e.g. Katsuwonus pelamis) [1], which means that identification of the fish species is important to avoid fraudulent and vague labeling of canned tuna [7].

36: Also, two-dimensional electrophoresis and mass spectrometric techniques were used, please add them and cite:

Pepe T., Ceruso M., Carpentieri A., Ventrone I., Amoresano A., Anastasio A. (2010) “Proteomic analysis for the identification of Thunnus genus three species” Vet Res Commun, 34 (Suppl 1):S153–S155

Response: As you recommended, we added reference [14] and revised the sentence in line 39.

L 57-67: Please, specify how did you proceed with the species recognition and who did it (morphological? From expert?)

Response: The five species of tuna, Longtooth grouper, convict grouper, marlin, and sailfish were obtained from the Ministry of Food and Drug Safety (Osong, Korea) as reference samples. 11 nontarget fish species were purchased from the online store and morphologically confirmed through the image database of the National Institute of Fisheries Science.

As you recommended, we revised the sentence in lines 63-73 as follows:

Lines 63-72: The five species of tuna: Bigeye (T. obesus), skipjack (K. pelamis), Atlantic bluefin (T. thynnus), albacore (T. alalunga), yellowfin (T. albacares) and longtooth grouper (Epinephelus bruneus), convict grouper (Epinephelus septemfasciatus), marlin (Tetrapturus angustirostris), and sailfish (Istiophorus platypterus) were obtained from the Ministry of Food and Drug Safety (Osong, Korea) as reference samples. Other 11 nontarget seafood (fish) products such as common carp (Cyprinus carpio), goldfish (Carassius auratus), Chinese muddy loach (Misgurnus mizolepis), snakehead (Channa argus), Pacific cod (Gadus macrocephalus), Alaska pollock (G. chalcogrammus), Nile tilapia (Oreochromis niloticus), Pacific saury (Cololabis saira), Pacific chub mackerel (Scomber japonicus), Atlantic salmon (Salmo salar), and masu salmon (Oncorhynchus masou) were purchased from the online store and morphologically confirmed through the image database of the National Institute of Fisheries Science.

Figure 3: Please, specify which is “P” in the legend

Response: As you recommended, we specified "P" to positive control in the legend of figure 3 as follows:

Figure 3, line 164: lane P, positive control (10 ng of DNA from target species)

Reviewer 4 Report

The method developed by the authors seems to be a very useful tool in the official diagnosis of food adulteration. The presented method shows its possibilities and research results in an accessible way. This is a good reason for the article to come out. Personally, I miss the detailed data on specificity, accuracy and sensitivity calculated in accordance with ISO standards in the study. As far as possible, I ask the authors to consider the possibility of adding this data in the article, it would significantly increase the area of ​​interest in the article and use in practice.
In addition, I did not find information about the validation of the method for heat-treated products, only information in tab. 2 about their application. I understand that the authors previously performed such tests, for example in the form of intra-laboratory verification? The results of such verification would support the relevance of the article.
Literature cited as 23 rather does not fully correspond to the content of the article.
line 77 LOD the abbreviation has no expansion.
In the section on DNA extraction, I do not understand why the authors referred to the method used for extraction from fruits and vegetables. Since in 2007 Chapela et al. Made a Comparison of DNA extraction methods from the muscle of canned tuna for species identification.
Also in the case of isofocusing proteins in the PH gradient, there are more recent publications showing the possibility of combining SDS techniques with isofocusing in the identification of fish species.

Author Response

The method developed by the authors seems to be a very useful tool in the official diagnosis of food adulteration. The presented method shows its possibilities and research results in an accessible way. This is a good reason for the article to come out. Personally, I miss the detailed data on specificity, accuracy and sensitivity calculated in accordance with ISO standards in the study. As far as possible, I ask the authors to consider the possibility of adding this data in the article, it would significantly increase the area of ​​interest in the article and use in practice.

Response: As you recommended, we added the detailed data on specificity, accuracy and sensitivity calculated in accordance with ISO standards (lines 182-183; Supplementary Table 1).

Lines 182-183: The detailed data on specificity, accuracy and sensitivity calculated in accordance with ISO standards are shown in Table S1.

In the ISO 20395:2019(en), specificity is expressed as a percentage, calculated as 100 × the number of true negative values (TN) divided by the sum of the number of true negative plus the number of false positive values (FP), or 100 × TN / (TN + FP).

Accuracy is affected by the prevalence of the target condition. With the same sensitivity and specificity, accuracy of a particular examination procedure is 100 × (TP + TN) / (TP + TN + FP + FN).

Sensitivity is expressed as a percentage, calculated as 100 × the number of true positive values (TP) divided by the sum of the number of true positive values (TP) plus the number of false negative values (FN), or 100 × TP / (TP + FN).

TP: the number of targets as the number of true positive values

TN: the number of non-targets as the number of true negative values.

FP: the number of false positive values

FN: the number of false negative values

In addition, I did not find information about the validation of the method for heat-treated products, only information in tab. 2 about their application I understand that the authors previously performed such tests, for example in the form of intra-laboratory verification? The results of such verification would support the relevance of the article.

Response: Since canned tunas are usually manufactured by heating at a temperature of 100°C or higher, canned tunas in Table 2 refer to heat-treated products. As the intra-laboratory verification, the results of three experimenters mean '+++' indicated in Table 2. '+' means amplified at the target species, and all three experimenters amplified the target species and indicated as '+++’.

Additionally, we revised Table 2 legend in line 210 as follows:

Table 2, line 210: Application and intra-laboratory validation results of the multiplex PCR assay to commercial tuna products.

Literature cited as 23 rather does not fully correspond to the content of the article.

Response: We deleted reference #23 in text and references section.

line 77 LOD the abbreviation has no expansion.

Response: We revised ‘LOD’ to ‘limit of detection (LOD)’ in line 77.

In the section on DNA extraction, I do not understand why the authors referred to the method used for extraction from fruits and vegetables. Since in 2007 Chapela et al. Made a Comparison of DNA extraction methods from the muscle of canned tuna for species identification.

Response: Thank you for your comment. In a previous study, DNA was efficiently extracted from fruits and vegetables processed by the CTAB method [32] in our laboratory. Since this CTAB method was used in this tuna study, reference was cited. In addition, the reference suggested by reviewer was also used as CTAB method [33] in this study and was added to the references

As you recommended, we added reference [33] and revised the sentence in line 83 and references section as follows:

Line 83: Then, DNA was extracted from the pretreated canned tuna using the cetyl trimethyl ammonium bromide method [32,33].

Also, in the case of isofocusing proteins in the PH gradient, there are more recent publications showing the possibility of combining SDS techniques with isofocusing in the identification of fish species.

Response: As you recommended, we added more recent references showing the possibility of combining SDS techniques with isofocusing in the identification of fish species [8,9] and revised the sentence in lines 37-39 as follows:

Lines 37-39: Protein-based assays for fish species identification have been previously reported, such as isoelectric focusing, isoelectric focusing in immobilized pH gradients, two-dimensional electrophoresis, and enzyme-linked immunosorbent assay [8-15].